# NETworking for Health and in Disease: Neutrophil Extracellular Traps in Pediatric Surgical Care

**DOI:** 10.3390/children11030295

**Published:** 2024-03-01

**Authors:** Maximilian Dölling, Martin Herrmann, Michael Boettcher

**Affiliations:** 1Department of General, Visceral, Vascular and Transplant Surgery, University Hospital Magdeburg, 39120 Magdeburg, Germany; maximilian.doelling@med.ovgu.de; 2Department of Internal Medicine 3, Rheumatology and Immunology, Friedrich-Alexander-Universität Erlangen-Nürnberg (FAU), Universitätsklinikum Erlangen, 91054 Erlangen, Germany; 3Deutsches Zentrum für Immuntherapie (DZI), Friedrich-Alexander-Universität Erlangen-Nürnberg, Universitätsklinikum Erlangen (FAU), 91054 Erlangen, Germany; 4Department of Pediatric Surgery, University Medical Center Mannheim, Heidelberg University, 68167 Mannheim, Germany; michael.boettcher@medma.uni-heidelberg.de

**Keywords:** extracellular traps, thromboinflammation, diabetes mellitus, tissue adhesions, wound healing, appendicitis, necrotizing enterocolitis, burns, gallstones, tumor microenvironment

## Abstract

This comprehensive review examines the role of Neutrophil Extracellular Traps (NETs) in pediatric surgery. Focusing on NET formation, functions, and implications, this study highlights their dual impact in infection control and contribution to tissue damage after surgery. It covers the role of NET formation in a range of pediatric conditions including immunothrombosis, formation of peritoneal adhesions, appendicitis, burns, gallstones, tumors, and necrotizing enterocolitis (NEC). The results underscore the significance of NETs in fighting infections and their association with complications like sepsis and delayed wound healing. The breakdown products of NETs as a diagnostic tool of the clinical course of acute appendicitis will also be discussed. Understanding NET formation in the pathophysiology can potentially help to find new therapeutic approaches such as the application of DNase and elastase inhibitors to change the clinical course of various diseases in pediatric surgery such as improvement of wound healing, adhesion formation, NEC, and many more.

## 1. Introduction

Neutrophils, the most populous cellular components of the innate immune system, perform critical surveillance functions in the bloodstream, in tissue, and on the external as well as internal body surfaces. Their ability to patrol various compartments and to respond to infectious agents underscores their importance in maintaining tissue homeostasis [1]. Initially, prevailing scientific understanding posited that the lifespan of non-activated neutrophils ranged between 1.5 to 8 h in both mice and humans [2]. However, contemporary research has revealed a markedly extended and variable lifespan for these cells. They display survival durations reaching up to 5.4 days, further amplified upon activation by cytokines [2]. The latter significantly prolong the survival period of neutrophils at the site of inflammation [2]. Neutrophils employ various mechanisms to combat invading pathogens, including phagocytosis, the release of anti-microbial peptides, and the generation of reactive oxygen species [3]. Takei et al. first described neutrophils to decondense their chromatin content followed by rupture of the nuclear envelop and release of their chromatin into the extracellular space after administration of phorbol myristate acetate (PMA) in vitro [4]. In 2004, this process has been assigned to as a neutrophil-mediated defense mechanism against invading pathogens due to the anti-microbial nature of extracellular chromatin and adherent enzymes and, therefore, named Neutrophil Extracellular Traps (NETs) by Brinkmann et al. [5]. Since then, many studies were performed to assess the mechanism of NET formation and their role in health and disease. Besides their classical, beneficial function, NETs have a dual nature and can be both beneficial and harmful in conditions of disease [6]. This duality arises from dysregulation of homeostatic processes, involving unregulated release or delayed clearance of NETs [7]. Summarized by Knopf et al., NETs have been identified to be crucial in many pathologic conditions such as (auto-)immunity, sepsis, thrombosis, wound healing, pancreatitis, lithopathies and malignancy, all of which are also relevant to pediatric surgical patients [6,7,8,9,10,11,12]. To understand the implications of NETs in surgery, it is essential to recap the process of NET formation. Various stimuli have been identified to induce NET formation. These include components from both Gram-positive and Gram-negative bacteria, fungi, protozoa, and their respective products [5,13,14]. Additionally, alterations in the microenvironment, such as bicarbonate concentration and alkalosis, have been implicated in the promotion of NET release [15]. Conversely, hypoxia inhibits NET formation by the hyperexpression of HIF-1α [16]. Interleukins, particularly IL-6 and IL-8, are also recognized as NET inducers [17,18]. Receptors on the plasma membrane such as toll-like receptor 2 and 4, Fc-receptors, and complement receptors initiate signaling cascades that lead to the activation of protein-arginine deaminase type 4 (PAD4) [19]. PAD4 activity results in citrullination of a plethora of target proteins and is key for chromatin decondensation, which is followed by ruptures of the granula, nuclear envelope, and eventually, the plasma membrane [20]. Consequently, neutrophils release decondensed DNA into the extracellular space as NETs [4,5]. The latter primarily consist of chromatin and histones aligned in a web-like structure decorated with numerous granula-derived enzymes, including neutrophil elastase, myeloperoxidase, and cathepsin G and many more. These enzymes display anti-microbial and cytotoxic activities [5,21,22]. Accumulating NETs form aggregates (aggNETs), which entrap and kill bacteria [23]. However, being decorated with a high concentration of proteolytic enzymes, they also limit inflammation and tissue damage, thereby restoring homeostasis [24,25]. Dysregulation of NET formation can result in disease being relevant to pediatric surgery (Figure 1).

## 2. Perioperative Infections

Neutrophils exposed to pathogenic bacteria, such as Staphylococcus aureus and Shigella spp, release NETs [5]. These directly interact with the bacteria and reduce their virulence factors in laboratory settings [5]. The mechanism is as follows: Tissue-invading activated neutrophils are triggered by bacterial virulence factors to release NETs [5]. NETs ensnare invading pathogens and kill them using their active antimicrobial components, which increases inflammation [5]. The importance of NETs was shown in experimental settings, as targeting NETs with monoclonal antibodies abrogated the killing of Staphylococcus aureus [26,27]. In high concentrations, NETs can accumulate and form aggregated NETs (aggNETs). These web-like structures have the ability to physically entrap invading pathogens. Due to the high concentration of anti-microbial compounds and the assistance of other immune cells and complement, invading pathogens are controlled and eventually killed. In vitro models where neutrophils were deprived of PAD4, an essential enzyme for chromatin decondensation, or where NETs were broken down by DNases, have demonstrated that in certain settings NET formation is required for bacterial killing by neutrophils [28]. While NETs play a beneficial role in the initial bacterial defense, emerging studies have shed light on a dual aspect of NET function in severe polymicrobial sepsis [21]. In a murine abdominal sepsis model, the administration of DNase has resulted in increased systemic dissemination of microbes [29]. Furthermore, in a further murine sepsis model, infant mice had more severe sepsis and higher levels of NETs compared to adult controls [30]. Interestingly, some bacteria may be able to evade clearance when being trapped within NET and it appears that the application of DNAse-1 improves the effectiveness of antibiotics in a murine sepsis with improved survival of mice compared to DNAse treatment alone [31,32]. Moreover, excessive NET formation has been linked to collateral damage in vital organs such as the liver, lungs, and kidneys during severe sepsis due to cytotoxic activity of histones, hypercoagulability, and by immunothrombosis [21,33,34]. While NETs contribute the prevention of the spread of bacterial infections, dysregulated NET formation in severe postoperative polymicrobial sepsis can precipitate life-threatening coagulopathies and proteolytic damage of the extracellular matrix.

## 3. Immunothrombosis

The term “immunothrombosis” has been coined to emphasize the link between innate immunity, coagulation, and thrombus formation [35]. This phenomenon has evolved as a defense mechanism, aiming to physically confine infectious agents within occluded blood vessels. This limits pathogen dissemination via circulation and constrains inflammatory responses [33]. However, in several pathological conditions, including sepsis-associated coagulopathy [9], necroinflammation [36], severe COVID-19 [37,38], and ischemia-reperfusion injury, an exaggerated form of immunothrombosis exacerbates the disease [39]. Beyond the role of NETs in physically trapping microbes, immunothrombosis serves hemostatic functions during mucosal damage, such as in acute exacerbations of ulcerative colitis [40]. Neutrophils and components of NETs interact with platelets, serine proteases in the coagulation cascade, components involved in fibrinolysis, and the fibrin mesh itself. These interactions occur through membrane-bound receptors, released effector proteins, chromatin present within NETs and extracellular vesicles [41]. Also, activated platelets can stimulate neutrophils to release NETs, while platelets can also bind to and aggregate on extracellular chromatin [9]. Thus, aggNETs serve as a framework for thrombogenesis, leading to obstructions of vessels and ducts [10,42]. PAD4 has been identified as key player linking inflammation and thrombosis. In vivo experiments showed that the injection of human PAD4 stimulated the production of von Willebrand factor (vWF) complexed with platelets in mesenteric venules. These complexes, typically degraded by ADAMTS13, become resistant when ADAMTS13 undergoes citrullination. This post-transcriptional modification significantly reduced its enzymatic activity [43]. Citrullination, e.g., by PAD4, also inhibits the anti-coagulative anti-thrombin, C1INH, 1-anti-plasmin, and PAI1/PAI2. This canceled the inhibition of the serine proteases thrombin, plasmin, and tissue plasminogen activator within the thrombo-inflammatory microenvironment [44,45]. PAD4-mediated thrombin activation has the ability to further modify thrombi and their growth via FXIII-mediated cross-linking [40]. Thombosis of the microvascular coupled with endothelial dysfunction induced by NETs, reduced capillary blood flow, contributed to multi-organ failure, and increased mortality [46,47]. Consequently, aggNETs drive the pathogenesis of various pathologies, such as the occlusions of pulmonary blood vessels observed in patients with COVID-19, coronary vessels during acute myocardial infarction, the development of atherosclerosis, and cerebral vessels in ischemic stroke [37,48,49,50]. Due to tissue trauma, increased levels of NETs are peri- and postoperatively released into the blood stream. The latter was associated with higher rates of postoperative thrombosis [33,35,41,51]

## 4. Wound Healing

Wound healing is a crucial process in the postoperative period, but it can be fraught with complications leading to significant morbidity and mortality, as well as a tremendous health care expenses with estimated costs of more than USD 3 billion per year in the US [52]. As a first reaction to trauma, the innate immune system orchestrates wound healing by the exhibition of pro- and anti-inflammatory properties [53]. Neutrophils are the first immune cells to migrate to the wound site, constituting the predominant cellular component during the skin’s reparative process [54,55]. Their primary function is to phagocytose and eliminate invading pathogens and cell debris [3]. In murine experiments, mice depleted of neutrophils under sterile conditions displayed an unexpected outcome: the absence of neutrophils led to expedited epidermal wound closure [56]. This phenomenon extends to diabetic mice with neutropenia, as they exhibited remarkable rates of wound closure in comparison to their non-neutropenic counterparts [18]. However, NETs have a dual role in wound healing. On the one hand, NETs beneficially combat invading pathogens, prevent their dissemination, and resolve inflammation, thereby limiting infection as an important risk factor [57]. On the other hand, Wong et al. demonstrated a predisposition of neutrophils in diabetic mice to form NETs, with wounds exhibiting significantly higher levels of NETs in comparison to those in non-diabetic mice [17,57]. This increase in NETs was found to be dependent on PAD4 and leads to elevated extracellular levels of NET-bound histones at the wound site [57]. Intriguingly, the digestion of NETs utilizing DNase1 resulted in better wound healing in both diabetic and non-diabetic mice. This indicates a significant role of NETs in impeding the wound healing process [57]. PAD4-knockout mice not only displayed accelerated wound healing, but also demonstrated faster re-epithelialization compared to controls [57]. Moreover, PAD4-knockout mice with diabetes had no impairment in wound healing. These compelling findings were further corroborated by a study on diabetic ulcers in humans, which revealed overexpression of NET components, specifically extracellular DNA (ecDNA), histones, and neutrophil elastase levels in non-healing diabetic wounds. Heuer et al. assessed primary and secondary wound healing after surgical laparotomy and burn wounds in a murine model with wildtype, PAD4-knockout, and DNAse-knockout mice. They observed that DNase1 improved collagen I and III deposition in the process of scar formation in mice after laparotomy. After thermal injury, the times of wound closure were shorter after DNase1 treatment as well as in PAD4-knockout mice [57]. The results of former experimental work suggest that these molecules and the process of NET formation represent pivotal therapeutic targets for enhancing wound healing across a spectrum of clinical scenarios.

## 5. Peritoneal Adhesions

Peritoneal adhesions occur in 54% of patients following abdominal surgeries [58]. Five years after open abdominal surgery (excluding appendectomies), 5.3% of child patients were readmitted because of adhesion-related symptoms, with the highest risk of readmission after the formation or closure of ileostomy, at 25% [59]. They are the leading cause for late complications such as intestinal obstruction [60,61]. Additionally, adhesions might lead to unspecific chronic abdominal pain, fertility issues and are a risk factor for collateral damage to other organs in relaparotomy with adhesiolysis (6%). They display a major effect on the quality of life and an enormous burden to health care systems [60,62,63]. Adhesion formation is a vital aspect of peritoneal healing and encompasses processes such as hemostasis, angiogenesis, and tissue remodeling [64]. In this context, inflammation orchestrated by the innate immune system, particularly through the recruitment of neutrophils, plays a pivotal role in adhesiogenesis [52]. A recent study has unveiled the presence of fibrin–NET complexes within both murine and human adhesions, with an example displayed in Figure 2 [64]. Intraperitoneal ecDNA was observed to peak 72 h after induction of the adhesions. Intriguingly, the use of DNases to degrade ecDNA effectively prevented the formation of adhesions induced by laparotomy in a murine model [64]. Additionally, PAD4-knockout mice exhibited reduced adhesion formation [64]. Conversely, the DNase1-knockout mice developed substantially more collagen deposits I and III compared to wildtype mice. These findings suggest that NETs serve as the initial scaffold, and fibrin and later collagen attachment functions to stabilize the primary structure, ultimately leading to the development of mechanically robust adhesions [64]. However, the trigger for adhesion formation is still elusive and needs further investigation.

## 6. Acute Appendicitis

Acute appendicitis (AA), the acute inflammation of vermiform appendix, is the most common atraumatic abdominal surgery emergency in children older than 2 years of age, with a lifetime risk of 6.7–8.6% in women and men, respectively [65,66]. AA peak incidence is in early adolescence between 10 and 19 years of age [67]. Although AA is the most common cause for emergency surgery in children, its underlying pathophysiology remains poorly understood [68]. The concept of three distinct forms of appendicitis, each with its own pathophysiological basis, has been proposed, largely based on studies from adults. The first type is thought to result from the obstruction of the appendicular lumen by fecoliths/appendicoliths [68,69]. The second type is caused by viral infections as summarized by Soltani et al. [70]. Both entities share mucosal ulcerations and subsequent secondary bacterial invasion with acute inflammatory responses. These include the infiltration of neutrophils into mucosal and submucosal layers [71]. The depth of neutrophil infiltration correlates with advanced stages of the disease [69]. Indeed, meta-analysis of histopathological studies showed a high neutrophil–lymphocyte ratio to be predictive for the presence and clinical stage of AA [72]. Interestingly, the third and rarest type of “neurogenic appendicitis” does not show significant neutrophil infiltration [73]. Clinically, conservative treatment of non-complicated appendicitis has been proposed in recent years. However, the identification of patients with non-complicated AA is still challenging. As CT scans involve radiation and are generally avoided for pediatric diagnostics [74], the prediction of complicated AA typically relies on patient history, clinical findings, laboratory values such as WBC count, granulocyte count, CRP concentration, and ultrasound [75,76]. Many efforts have been made to find new biomarkers to help differentiate between complicated and uncomplicated appendicitis [77]. In nearly all cases of acute appendicitis, infiltrating neutrophils release NETs [5]. In a pilot study of a murine model of AA and children, the author observed that the number of NETs in tissue correlated with the severity of the disease [78]. The breakdown products of NETs such as DNA-NE-, DNA-MPO-complexes, cell-free DNA and citrullinated Histone (citH3) were increased in AA and correlated with tissue concentrations of NETs and the severity of the disease [78]. A large study involving 198 children with 133 with histological AA confirmed these results and identified citH3 and cell-free DNA as reliable predictors for AA [79]. DNA-Myeloperoxidase (MPO) complexes and citH3 excel as diagnostic markers for complicated AA. This provided valuable insights into outcomes such as duration of hospitalization, wound infections, abscess formation, and the overall rates of complication. It outperformed common markers such as leucocyte count and CRP [79]. Even though NETs are observed in most, if not all cases of AA, their exact role in the pathophysiology is still elusive.

## 7. Necrotizing Enterocolitis

Necrotizing enterocolitis (NEC) is associated with high mortality and morbidity in preterm infants. The exact mechanisms behind NEC are still elusive. Research suggests key factors in its development: prematurity and formula feeding [80,81]. In premature infants, the underdeveloped mucosal barrier combined with increased toll-like receptor 4 levels in the gut lining led to a hyper-reactive gastrointestinal response, a surge in inflammatory cytokines and chemokines, augmented leukocyte migration, destruction of intestinal epithelial cells, compromised gut barrier function, and the abnormal movement of bacteria from the gut lumen into surrounding tissues [82,83,84,85]. The role of NETs in NEC has recently been reviewed by Klinke et al. [86]. NETs were found in various tissues, as well as in the patients’ sera, stool samples of infants, as well as mice afflicted by NEC [87,88,89]. McQueen et al. unveiled a link between calprotectin, an enzyme from neutrophil granula, and NETs in the intestinal tissues of infants with NEC. This suggests calprotectin release due to activated neutrophils and NET formation in the intestinal tissue [90]. Premature infants with NEC and consecutive bacteremia with signs of organ dysfunction showed elevated levels of cell-free DNA, an indicator denoting the presence of circulating NET degradation products, when compared to control subjects [88]. Similarly, heightened levels of nucleosomes, another marker of NET release, were detected in the blood of newborns with NEC stage II and beyond, compared to controls matched by gestational age [88]. In a murine model that employs intermittent hypoxia, lipopolysaccharides, and formula feeding, serum markers of neutrophil activation and NET release as well as NETs in histomorphometry correlated with severity and mortality of NEC [88]. Pharmacological inhibition of PAD4 with Cl-amidine downregulated the release of NETs and reduced NEC-related tissue damage, inflammation, and mortality. These results align with recent investigations indicating that the degradation of NETs by recombinant DNase1 considerably reduced intestinal inflammation [91]. Also, the inhibition of NET formation by gasdermin D polymerization using disulfiram improved organ function and survival in sepsis [92]. In summary, previous data suggest NETs are crucial for innate defense, particularly in the early clearance of bacteria and bacterial products in NEC. It is still elusive whether there is a causal relationship between NEC and NETs.

## 8. Burns

The immediate physiological responses following burn injuries are intricate and involve a series of overlapping phases of wound healing. These include hemostasis, inflammation, proliferation, and remodeling [93]. NETs prolonged the inflammatory phase, and activated neutrophils persist for months in burn patients following thermal injuries [57,94]. In an experimental model, Elrod et al. exposed mice to hot water. This elicited a systemic response targeting the structural integrity of dermal collagen fibers and initiated the process of burn healing [95]. In this experiment, neutrophils infiltrated the interstitial space between dermal white adipose tissue and the panniculus carnosus with consecutive apoptosis of the surrounding tissue. The systemic responses with the release of cytokines and chemokines resulted in a substantial upregulation of markers indicative of neutrophil activation and the formation of NETs. It was evident in plasma, wounds, and various organs, including lung and liver. In this study, activated neutrophils and NETs have been identified as causative factor for the distant damage to liver and lung tissue 24 h after burn injury. This is putatively due to oxidative stress and/or immunothrombosis. After thermal injury, DNase1L3, was detected in wound, liver, and lung samples. This enzyme acts as a regulator of NET metabolism [95]. In summary, dysregulation of NETs due to massive tissue damage may cause the systemic reaction and end organ damage after burn injury. NETs are, therefore, potential therapeutic targets for the treatment of severe burns.

## 9. Biliary Atresia

Biliary atresia (BA) represents a severe fibro-inflammatory disease marked by the obstruction of extrahepatic and intrahepatic bile ducts in neonates, posing a potentially lethal outcome if not addressed promptly [96]. As the leading cause of pediatric liver transplantation globally, the quest for novel therapeutic strategies to obviate the necessity for surgical intervention remains challenged by the disease’s intricacy and the current gaps in comprehending its pathogenesis [96]. Neutrophil accumulation has been documented in the vicinity of bile ducts in patients with BA [97]. For further studies, the sole established model of BA to study innate immunity, employing an infection with Rhesus Rotavirus Type A in newborn BALB/c mice, has facilitated the elucidation of crucial cellular and molecular mechanisms underpinning epithelial damage and ductal obstruction [98,99]. Using this model, Zhang et al. demonstrated that depleting neonatal Gr-1+ cells averts the onset of BA in a murine disease model [100]. In a consecutive study, they identified functionally activated neutrophils, called CD177 + cells, being the main population of Gr-1+ cells in murine liver with biliary atresia. Additionally, those cells were found to express interferon-stimulated and neutrophil degranulation genes [101]. Interestingly, CD177-knockout mice experienced a delayed onset of BA with reduced morbidity and mortality, respectively. Following experiments on CD177+ cells in cell cultures, increased apoptosis of biliary epithelial cells was recognized due to high levels ROS and NETs. Furthermore, N-acetylcysteine reduced CD177+ cells and ROS levels in a clinical pilot study, thus highlighting the potential therapeutic benefits of NET-directed therapies [101].

## 10. Traumatic Injuries

Traumatic events lead to the release of damage-associated molecular patterns (DAMPs) which leads to a systemic inflammatory response characterized by the activation of the innate immune system. In many traumatic conditions, NETs have been recognized within the sterile inflammatory milieu following injury. Decorated with proinflammatory enzymes such as MPO, NE, and MMP-9 and with cytotoxic histones, NETs contribute to tissue damage and inflammation [12]. After orthopedic traumatic injury, NET biomarkers are increased in peripheral blood compared to in patients who received elective hip replacement [102]. Patients who received DNase treatment after major trauma had reduced NET biomarkers in peripheral blood [103,104].

A special condition is traumatic injury of the spinal cord, which is a serious condition with high impact to the individual and leads to high costs for the health care system. The pathophysiology of traumatic spinal cord injuries is sequenced into two phases: (1) primary tissue injuries occur due to mechanical stress, which disintegrates neurons from axons, glia cells, and the blood–brain barrier. Attracted by chemotaxis, activated neutrophils infiltrate the site of injury, release NETs, and thereby increase inflammation and initial tissue damage due to their cytotoxic compounds [105]. (2) Secondary injuries develop after additional vascular damage and edema formation [106]. In a murine model of spinal cord injury, pharmacological inhibition of phosphodiesterase 4 (PDE4) led to decreased neutrophil infiltration and reduced MPO concentration at the site of injury, resulting in less apoptosis due to ROS release and better locomotor outcome in mice [107,108]. In another study on mice, NET formation was associated with cerebral hypoperfusion and tissue hypoxia [109]. Additionally, intravenous DNase1 treatment of rats reduced pro-inflammatory cytokine concentrations in peripheral blood and reduced neuroinflammation, edema, and fibrosis [105]. In mice with traumatic brain injury, DNaseI stabilized the blood–brain barrier, reduced brain edema, and secondary brain injury [110]. Despite the insights gained from murine models, the role of NETs in the pathophysiology of traumatic injuries remains unclear, warranting increased attention in future research endeavors.

## 11. Gallstones

Even though gallstones are more prevalent in adults with hyperalimentation, they are also associated with conditions such as hemolytic anemia, congenital biliary diseases, leukemias, short bowel syndrome, and exposure to total parenteral nutrition or antibiotics in children [111,112]. Gallstones exhibit geographical variations in type, with cholesterol stones being predominant in Western countries [113]. A common factor in the formation of all types of gallstones is the crystallization and precipitation within the gallbladder as calcium and cholesterol salts [114]. NETs have been shown to facilitate the aggregation of calcium and cholesterol crystals, thus contributing to the assembly of gallstones [115]. Neutrophils continually survey the body in homeostasis, allowing them to access the bile ducts [99]. During inflammatory responses, fully activated neutrophils tend to accumulate in the liver, allowing them access to the bile ducts [98]. The process of NET aggregation reportedly orchestrates the compact packing of uric acid crystals during the formation of gouty tophi [23,116,117]. A similar mechanism may work during gallstone formation. Here, ecDNA released by neutrophils facilitates the clustering of calcium and cholesterol crystals. Inhibition of NET formation in Ncf1** mice or reduced chromatin decondensation in PAD4-knockout mice reduced the prevalence and size of gallstones induced by a lithogenic diet [11]. PADI4 inhibitor or metoprolol, a selective β1-adrenergic receptor antagonist known to dampen neutrophil activity, also reduced the growth of gallstones [11]. Gallstones display high neutrophil elastase activity on their surfaces and are covered by abundant extracellular chromatin. This suggests that early stones induce recruitment of further neutrophils and NET formation on their surfaces. In addition, chemo-attractive factors, like complement cleavage products, can be activated by cholesterol crystals and enhance neutrophil influx. This perpetuates the growth of the gallstone, accompanied by the continued packaging and stabilization of layers of calcium and cholesterol crystals. Other triggers that mobilize neutrophils to enter the biliary ducts, such as ascending bacterial infections, may further aggravate the growth of gallstones. Thus, NETs seem to be the driving factor of gallstone growth. However, not all stone entities have been tested yet.

## 12. Tumors

While cancers like Wilms tumor and leukemias occur in children, the prevalence of tumors is generally higher in adults. As such, our understanding of immune responses to tumors in pediatric populations may be enhanced by first examining insights gained from adult tumor studies. This approach involves translating knowledge of tumor immunology from adults to better understand and address pediatric cancers. Increased prevalence of NETs in tumors is associated with a poorer prognosis among cancer patients. This is reflected by higher histopathological tumor grades, higher rate of disease progression and metastasis, and reduced disease-free and cancer-related survival across various cancer types [118,119,120,121]. The detection of NETs in cancer patients often involves measuring elevated serum levels of MPO-DNA complexes [120,122,123], and histomorphometry within tumor tissues [118,124]. Recently, a correlation between NETs in tissues of human colorectal carcinoma and UICC stages 3–4 has been reported [118]. In-depth investigations in murine models have shed light on the roles of NETs in tumorigenesis. Intriguingly, surgical stress and increased levels of lipopolysaccharides following postoperative infections have been found to induce the release of NETs concurrently with an increased development of metastases [120,123]. NETs may directly facilitate the development of metastasis by capturing tumor cells at distant sites by interactions between NETs and the coiled-coil domain-containing protein 25 (CCDC25). The latter is expressed on cancer cells of colorectal, breast, prostate, and liver cancers [121]. Moreover, NETs co-regulate several crucial processes involved in cancer development and metastasis including the immune-evasive microenvironment, the activation of dormant tumor cells, tumor cell migration into surrounding tissues, angiogenesis, and increased vascular permeability [125,126]. Furthermore, NETs induce a mesenchymal, pro-metastatic phenotype in various cancer cell lines, including breast, colorectal, gastric, and pancreatic cancers. They trigger epithelial–mesenchymal transition, resulting in enhanced tumor cell migration and invasion [118]. In summary, current knowledge strongly suggests that NETs play active roles in cancer progression and may represent potential targets for therapeutic interventions and preventive strategies.

## 13. Therapeutic Implications

Even though NETs are integrally involved in various aspects of surgical practice, the incorporation of anti-NET strategies in surgical methodologies has been minimally explored. This might be because of their sensitive homeostatic processes and their dual role with their benefits or harm depending on the situation and the time course of disease. Rash inhibition of NET formation may reduce the ability of the immune system to fight infectious agents and limit inflammation. Also, depending on the stage and type of cancer, the disruption of NETs might lead to tumor progress. On the other hand, the promotion of NETs formation may lead to a higher amount of autoantigens, thereby increasing the risk of autoimmunity or the impairment of wound healing [12].

Nevertheless, there are efforts with preclinical studies that indicate that these anti-NET interventions are biologically safe and exert minimal impact on the immune response to pathogenic invasions. Notably, several pharmacological agents with anti-NET properties have already received regulatory approval for clinical use. Among these are recombinant DNases that cleave the DNA backbone of the NETs and, thus, promote NET metabolism [5,127]. This enzymatic treatment has demonstrated efficacy in cystic fibrosis management, primarily by reducing the viscosity of pulmonary secretions and consequently lowering infection risk [128]. Also, DNase1L3 has a great potential to cleave extracellular chromatin [129]. Another therapeutic approach targeting NETs involves the use of neutrophil elastase inhibitors. These function as serine protease antagonists, specifically targeting neutrophil elastase, a pivotal enzyme in NET formation [130]. CXCR2 inhibitors represent a novel and promising class of NET-targeted therapies. These inhibitors act on CXCR2, a recently identified regulator of NETs in COPD [131]. Clinical trials involving CXCR2 inhibitors have primarily focused on COPD, asthma, and tumors. Cl-amidine is known for the inhibition of PAD4, one of the key enzymes in NET formation, thereby reducing the release of NETs. A preclinical trial focused on the inhibitory function of Cl-amidine on NET formation and showed a reduction in thrombosis and atherosclerotic lesion areas in a murine model of atherosclerosis [49]. The production of ROS is a key step in suicidal NET formation [132]. Therefore, several studies focused on the inhibition of ROS production to reduce NET formation [132]. N-acetylcysteine is a mucine lysine and is widely used for the treatment of chronic obstructive pulmonary disease and bronchiectasis to reduce mucous viscosity [133,134]. Additionally, is has antioxidant effects and counteracts the toxic effects of reactive oxygen species (ROS) [132]. It was shown that N-acetylcysteine reduces thrombus formation in vivo, counteracting immunothrombosis by NET formation [135,136]. Other ROS scavengers such as methotrexate or Diphenyleneiodonium chloride have also shown potential to reduce NET formation in preclinical trials [137,138,139]. In summary, while preclinical trials have identified numerous methods to affect the release of NETs, no treatments targeting NET formation have been approved to date, resulting in limited data on human applications.

## 14. Conclusions

In the context of pediatric surgery, NETs are relevant due to their dual functionality. They are critical in combating infections by ensnaring and neutralizing pathogens, yet can induce tissue damage and exacerbate conditions like sepsis if uncontrolled. NETs play a role in wound healing and balance anti-microbial action against the potential for delayed recovery. NETs are implicated in the pathogenesis of peritoneal adhesions, appendicitis, immunothrombosis, occlusive diseases, stone diseases, and cancer. This illustrates their complex impact in various conditions of pediatric surgery. However, due to their dual role in the physiology of health and the pathophysiology of diseases, interventions in clinical courses are difficult to predict. This opens up a broad field for clinical research in the future. Therapeutic strategies targeting NETs, including DNase and elastase inhibitors, offer promising avenues to modulate the effects of NETs.

## Figures and Tables

**Figure 1 children-11-00295-f001:**
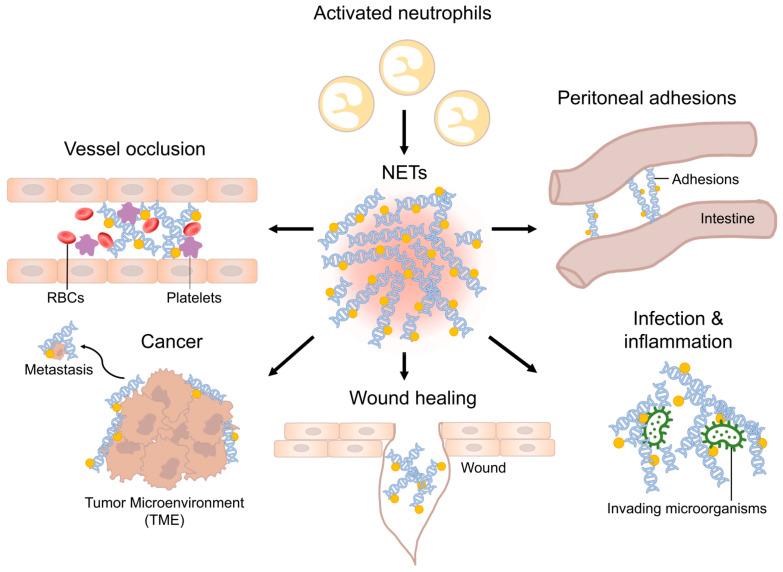
Schematic overview of NETs in various diseases. Neutrophils invade the site of inflammation and are activated. Activated neutrophils can perform NET formation. Dysregulation of NET homeostasis can lead to various diseases relevant to pediatric surgery.

**Figure 2 children-11-00295-f002:**
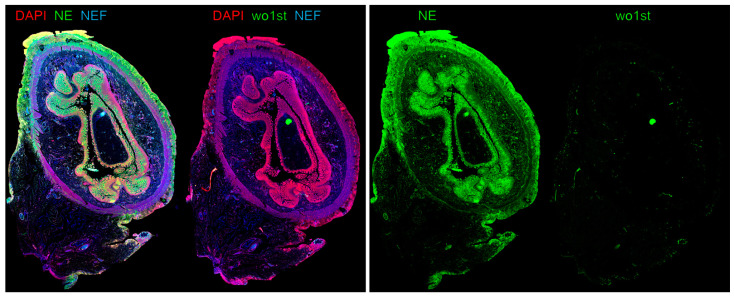
Immune fluorescence image of a transection of small intestine after acute perforation with neutrophil extracellular traps (NETs) forming peritoneal adhesions attached to the outer surface of the gut. Left: An overlay image shows the co-localization of neutrophil elastase (NE, green) and DNA (DAPI, red), indicating the presence of NETs on the peritoneal and mesenchymal layers of the small intestine. Interestingly, vascular occlusions in submucosal and subserosal layers of the small intestine could be identified by native endogenous fluorescence (NEF) due to NEF properties of hemoglobin. Note the NEF signal being colocalized with NE, suggesting that vascular occlusions in the small intestine re-associated with substantial NET release in small blood vessels. Right: The secondary fluorescent-conjugated antibody binds specifically to the antibody against NE and shows almost no unspecific binding to human antigens compared to staining without primary antibody (wo1st). Figure created by M. Herrmann.

## Data Availability

Not applicable.

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
