# Peer review of "NETworking for Health and in Disease: Neutrophil Extracellular Traps in Pediatric Surgical Care"

_children, 2024, doi:10.3390/children11030295_

Round 1

Reviewer 1 Report

Comments and Suggestions for Authors

I read with interest the manuscript entitled "NETworking for Health and in Disease: Neutrophil Extracellular Traps in Pediatric Surgical Care – a Review."

I suggest removing "– a Review." from the title.

I suggest adding "narrative review" to the manuscript type.

The abstract is too general. I suggest that it be more comprehensive with concrete conclusions.

Also, I ask that keywords be from MeSH terms. Considering the comprehensiveness of the manuscript, please add more, up to a total of ten.

Where are figures 1 and 2 taken from?

I suggest that you touch on the history of NET research in the introductory part. Who were the pioneers of research and how did research proceed?

Reading the manuscript, in relation to the title, most of the mentioned articles refer to basic medical science and research on animal models. I suggest that the focus be on human research.

I suggest that the "Therapeutic implications" section be more comprehensive with far more references of interest.

I suggest that you also touch on potential future research on this topic, regardless of the domain of pediatric surgery. Although you emphasized the pediatric surgical aspect in the title, the manuscript is dominantly formed from basic researchs.

Ultimately, the children's surgical aspect has only been marginally touched upon. The manuscript is dominated by basic researchs with your review of potential effects on surgical conditions in childhood. Please include more research on humans.

Comments on the Quality of English Language

Moderate editing of English language required.

Author Response

Dear Reviewer,

please find our response to your valuable feedback attached.

Best regards,

M. Herrmann, M. Boettcher and M. Dölling

Reviewer 2 Report

Comments and Suggestions for Authors

Thank you for sending me this work for consideration. Excellent work. However, there are some sections that need to be corrected.

1-I think the sentence of 93-100% adhesions after abdominal surgery is a bit exaggerated. The rate of adhesions formed in a standard acute abdomen should not be the same as in intestinal perforation. Either change this sentence or give a reference.

2-In your reference, the incidence of adhesions is 62.9% (60). Didn't you read this reference?

3-Although this study is a study on pediatric surgery, there are only the words pediatric surgery in a single sentence in the introduction section of the study. Look, you have filled in the references of adult articles in the acute appendicitis section, which is one of the most important topics of pediatric surgery. Do not change the acute appendicitis section, but the references to the information in this section should be references to pediatric appendicitis. 66-69 references should change.

4-What is the role of NET's biliary atresia? ( Zhang R, Su L, Fu M, Wang Z, Tan L, Chen H, Lin Z, Tong Y, Ma S, Ye R, Zhao Z, Wang Z, Chen W, Yu J, Zhong W, Zeng J, Liu F, Chai C, Guan X, Liu T, Liang J, Zhu Y, Gu X, Zhang Y, Lui VCH, Full PKH, Lamb JR, Wen Z,  Chen Y, Xia H. CD177+ cells produce neutrophil extracellular traps that promote biliary atresia. J Hepatol. 2022 Nov; 77(5):1299-1310. DOI: 10.1016/J.Jhep.2022.06.015. Epub 2022 Jul 5th PMID: 35803543.)

5-What is the role of NET's the post-traumatic process? (Liu FC, Chuang YH, Tsai YF, Yu HP. Role of neutrophil extracellular traps following injury. Shock. 2014 Jun; 41(6):491-8. doi: 10.1097/SHK.00000000000000146. PMID: 24837201.)

Author Response

(The authors gave the same response as above.)

Round 2

Reviewer 1 Report

Comments and Suggestions for Authors

Thank you for the corrections, answers and clarifications.

Comments on the Quality of English Language

 Minor editing of English language required.